# Effect of justification of wife-beating on experiences of intimate partner violence among men and women in Uganda: A propensity-score matched analysis of the 2016 Demographic Health Survey data

**Damazo T. Kadengye** [ID] *, **Jonathan Izudi** [ID], **Elizabeth Kemigisha** [ID], **Sylvia Kiwuwa-Muyingo** [ID]

African Population and Health Research Center (APHRC), Nairobi, Kenya

* dkadengye@aphrc.org, dkadengye@gmail.com

**Data Availability Statement:** All data files are publicly available at http://www.dhsprogram.com.

## Abstract

### Introduction

In some communities, rationalization of men's controlling attitudes is associated with the justification of gender norms such as wife-beating as a method of correcting spouse behaviour. In this quasi-experimental study, we investigate the causal effects of the acceptability of gender norms justifying wife-beating on experiences of sexual, emotional, and physical intimate partner violence (IPV) among Ugandan men and women.

### Methods and materials

We analysed the 2016 Uganda Demographic and Health Survey data using propensity-score matching. The exposure variable is the acceptability of gender norms justifying wife-beating measured on a binary scale and the outcomes are the respondent's lifetime experiences of sexual, physical, and emotional IPV. We matched respondents who accepted gender norms justifying wife-beating with those that never through a 1:1 nearest-neighbour matching with a caliper to achieve comparability on selected covariates. We then estimated the causal effects of acceptability of gender norms justifying wife-beating on the study outcomes using a logistic regression model.

### Results

Results showed that a total of 4,821 (46.5%) out of 10,394 respondents reported that a husband is justified in beating his wife for specific reasons. Among these, the majority (3,774; 78.3%) were women compared to men (1,047; 21.7%). Overall, we found that men and women who accept gender norms justifying wife-beating are more likely to experience all three forms of IPV. In the sub-group analysis, men who justify wife-beating were more likely to experience emotional and physical IPV but not sexual IPV. However, women who justify wife-beating were more likely to experience all three forms of IPV.

**Funding:** The author(s) received no specific funding for this work.

**Competing interests:** The authors have declared that no competing interests exist.

## Conclusions

In conclusion, the acceptability of gender norms justifying wife-beating has a positive effect on experiences of different forms of IPV by men and women in Uganda. There is, therefore, a need for more research to study drivers for acceptance of gender norms justifying wife-beating to enable appropriate government agencies to put in place mechanisms to address the acceptability of gender norms justifying wife-beating at the societal level.

## Introduction

Intimate partner violence (IPV) harms the victim's physical, sexual or psychological health. IPV is a grand issue that reflects the unequal power dynamics created within the binary gender system and is often perpetrated by those with more physical, cultural, or social power and inflicted upon those without [1–3].

Globally, an estimated 30% of women report having ever experienced physical and/or sexual violence by an intimate partner in their lifetime; and this is highest among women in the WHO African, Eastern Mediterranean, and South-East Asia regions at 37% [4, 5]. The 2018 global estimates based on data from 2000–2018 indicate that the lifetime prevalence of physical and/or sexual IPV among ever-married/partnered women aged 15–49 years was highest among the least developed countries with Oceania at 37%, Southern Asia, 35% and Sub-Saharan Africa at 33% [6]. Although estimates for men are rarely reported in the literature, a recent analysis of data from six East African countries shows that the prevalence of physical, sexual, and emotional IPV against men is about half that of women [7].

Strong evidence exists that social norms about the roles and behavior of men and women contribute to an increased level of IPV in various low- and middle-income countries [3, 4, 8, 9]. Given the importance of social norms such as male masculinity and female subordination in shaping acceptable behavior within communities, several programmes aimed at shifting norms and behavior around IPV have been implemented across Africa [10]. For example, the SASA! Project, a community mobilization project designed to transform gender relations and power dynamics, and reduce the social acceptability of IPV has been implemented in over 15 countries, including Uganda [10–14]. The general strategy of the SASA! project and several of its variants are to bring about changes in social norms through local activism, media, use of communication materials and community-based edutainment/theatre, and advocacy to reduce the social acceptability of IPV [10, 14]. In addition, Uganda has provided a favorable policy environment including the introduction of the Domestic Violence Act of 2010 to protect both men and women against gender-based violence [15].

Yet, the 2016 Uganda Demographic Health Survey (UDHS) shows that IPV prevalence remains very high, with 56% of ever-married women and 44% of ever-married men reporting to have ever experienced physical, sexual, or emotional violence by their current or most recent spouse/partner [16]. Furthermore, about four in 10 women and men (both 39%) experienced IPV in the 12 months preceding the survey [16]. In general, the prevalence of the different forms of IPV (physical, sexual, or emotional) has generally remained high in Uganda over the last decade [7]. Notably, there has been a stable and an unacceptably high prevalence of lifetime sexual IPV among ever-married Ugandan women, with estimates of 24.8% in 2006, 28% in 2011, and 22% in 2016 [16–18]. This could be a consequence of women's perceived low status in many societies and men's dominance over women which is enforced through sexual and physical violence [19]. The close association of violence with masculinity has created a

dangerous and unjust power dynamic that manifests in forms of violent physical, emotional, or psychological aggression and affects an alarming proportion of the population [2]. In some communities rationalization of men's controlling attitudes is associated with the justification of gender norms such as wife-beating [20], which may increase the risk of experiencing IPV [21, 22].

In this paper, we investigate the causal effects of the acceptability of gender norms justifying wife-beating on the experiences of sexual, emotional, or physical violence among Ugandan men and women using a quasi-experimental design with propensity-score matched (PSM) analysis. We use data from the 2016 Uganda Demographic and Health Survey (UDHS) data.

## Methods and materials

### Data source and study population

We analyzed data from a nationally representative population-based household survey, the 2016 UDHS [16]. Data collection took place from 20 June to 16 December 2016. Inner City Fund provided technical assistance through the DHS Program, which is funded by the United States Agency for International Development (USAID) and offers financial support and technical assistance for population and health surveys in different countries worldwide.

The survey sample was stratified and selected in two stages. The first stage consisted of the selection of 697 enumeration areas: 162 urban and 535 rural. Due to land disputes, one cluster from the Acholi sub-region was excluded to ensure the safety of enumerators. The second stage involved sampling of households within the clusters. This was achieved through a listing of all households within each of the 696 accessible selected enumeration areas between April and October 2016, with some listings overlapping with fieldwork. The survey drew maps for each of the sampled clusters and then listed all the households but institutional living arrangements, namely army barracks, hospitals, police camps, and boarding schools were all excluded.

To minimize the task of household listing, each large enumeration area yielding more than 300 households selected for the survey was segmented, and one segment was selected for the survey with probability proportional to segment size, and the household listing was conducted within the segment. The implication is that a cluster suggests an enumeration area or a segment of an enumeration area. Overall, a representative sample that consisted of 20,880 households corresponding to 30 per enumeration area or a segment of the enumeration area was randomly selected for the survey. All women aged 15–49 who were either permanent residents of the selected households or visitors who stayed in the household the night before the survey were eligible to be interviewed. In one-third of the sampled households, all men aged 15–54, including both usual residents and visitors who stayed in the household the night before the interview, were eligible for individual interviews. Data were collected using four questionnaires: the household, women, men, and biomarker questionnaires. Detailed procedures on sampling and methodology are available in the 2016 UDHS report [16].

The 2016 UDHS included a domestic violence module in all sampled households. Following the World Health Organization's (WHO) guidelines on the ethical collection of information on domestic violence, only one eligible person per household was randomly selected for the module which was implemented in privacy. In two-thirds of households, one woman aged 15–49 was randomly selected to receive the domestic violence module as part of her interview. In the remaining one-third of the households, one man aged 15–54 was randomly selected to receive the domestic violence module as part of his interview. In total, 9,232 women aged 15–49 and 4,011 men aged 15–54 (3,758 men aged 15–49) responded to the domestic violence questions. One percent of eligible women and men could not be successfully interviewed with the module because of lack of privacy or other reasons.

## Study design

We employed a quasi-experimental design with propensity score matched (PSM) analysis using data from all respondents in the 2016 UDHS who responded to the domestic violence module. The rationale behind the use of PSM analysis is to achieve balance in participant characteristics between two groups [23], a group that reported gender norms justifying wife-beating as acceptable (exposed) versus a group that reported it as not acceptable (non-exposed group). Balancing of participant characteristics controls for selection bias and the confounding of the association between the exposure and the outcome [24], thus mimicking a randomized controlled trial (RCT) [25].

## Variables and measurements

**Exposure.** The exposure of interest was acceptance of gender norms justifying wife-beating, with the exposed group being respondents reporting that it is acceptable and the non-exposed group consisted of respondents that reported it is not acceptable. During the 2016 UDHS [16], both women and men were asked to report whether or not beating one's wife was justified under the following five circumstances: 1) Wife goes out without telling husband; 2) Wife neglects the children; 3) Wife argues with husband; 4) Wife refuses to have sex with husband; and, 5) Wife burns the food. The justification of wife-beating was computed as the percentage of all women and men aged 15–49 who agree that a husband is justified in beating his wife under the above five circumstances.

**Outcomes.** The 2016 UDHS [16] included life-time experiences of sexual, physical, and emotional violence with the most recent intimate partner, and we considered as our study outcomes. Respondents who answered 'yes' to any of the provided questions were considered to have experienced IPV. More specifically, violence committed by the current spouse/partner (for women and men currently married or in intimate relationships) and by the most recent spouse/partner (for women and men formerly married or in past intimate relationships) was measured by asking all women and men if their spouse/partner ever did the following to them:

- Physical IPV: Push you, shake you, or throw something at you; slap you; twist your arm or pull your hair; punch you with his/her fist or with something that could hurt you; kick you, drag you, or beat you up; try to choke you or burn you on purpose; or threaten or attack you with a knife, gun, or any other weapon.

- Sexual IPV: Physically force you to have sexual intercourse with him/her even when you did not want to, physically force you to perform any other sexual acts you did not want to, or force you with threats or in any other way to perform sexual acts you did not want to.

- Emotional IPV: Say or do something to humiliate you in front of others or threaten to hurt you.

**Covariates.** We extracted the background characteristics of each respondent to be included in the statistical analyses as covariates, namely sex (male or female), age group (15–19, 20–24, 25–29, 30–34, 35–39, 40–44, 45–49, and 50–54), level of education (none/no education, primary, secondary, and higher), marital status (never in a union, currently in a union, and formerly in a union), number of living children, wealth index (poorest, poorer, middle, richer, and richest), religion (no religion, Anglican, Catholic, Muslim, Seventh Day Adventist, Pentecostal, and others), and the 15 regions in Uganda (Kampala, Central 1, Central 2, Busoga, Bukedi, Bugishu, Teso, Karamoja, Lango, Acholi, West Nile, Bunyoro, Tooro, Ankole, and Kigezi). These covariates are known to either influence the outcome or the exposure of interest

and were selected as potential confounders as appropriate for the assumptions of strong ignorability of treatment assignment or unconfoundedness [26].

## Data analysis

All data management and analysis were done in Stata version 15 [27]. To measure the effect of acceptability of gender norms justifying wife-beating on experiences of sexual, physical, and emotional violence, we used PSM analysis to create a counterfactual, a comparison group that was identical to the comparison group [26]. We used a logit model to generate propensity scores by regressing the exposure on the matching covariates and assessed the initial balance in propensity scores and covariates between the exposed and non-exposed groups by splitting the sample into equally spaced intervals [23]. We used the Student's t-test within each interval to assess statistically significant differences in the average propensity scores between the groups. The intervals were divided further in instances where we observed statistically significant differences and then re-tested until such differences were removed. The degree of overlap of propensity scores and balance in covariates was checked using a propensity score graph.

We then matched the exposed and non-exposed respondents on similar propensity scores using several approaches: pair matching with and without replacement, nearest neighbor matching with and without a caliper, and kernel matching to select the most appropriate approach with balance [28]. In pair matching, we matched respondents in the exposed group with that in the non-exposed group regardless of the quality of the matches. In nearest neighbor matching without a caliper, the respondents were matched based on similar propensity scores regardless of the distance/width within which the matching was implemented. However, in nearest neighbor matching within a caliper, the respondents were matched within a distance/width known as a caliper computed as 20% of the standard deviation of the propensity score in a 1:1 ratio to prevent bias from distant matches [23]. In Kernel matching, the weighted average of all the respondents in the non-exposed group was used to construct the missing counterfactual outcome to enable the use of more data and produce fewer differences. Of all these approaches, we selected the most appropriate one based on the balance of all covariates across the exposed and non-exposed groups and the significant reduction in propensity score pseudo-R-square value. We assessed covariate balance after matching and considered covariates with a standardized mean difference (SMD) of less than 0.25 as balanced.

We saved the matched dataset and used it for outcome analysis, where we fitted a conditional logistic regression analysis for the outcomes taking into consideration the matched pairs. We reported the results odds ratios with a 95% confidence interval.

We checked the robustness of the results to unmeasured confounders and the analytic approach using the Mantel-Haenszel (MH) bounds approach proposed by Rosenbaum, with distant gamma values to achieve statistical significance or insignificance considered indicative of robust findings [29], using the Stata command "*mhbound*".

## Ethics statement

We conducted secondary data analysis using the 2016 Uganda Demographic Health Survey de-identified data that is publicly available and requires no ethical approvals. However, permission to use these data was obtained from the Demographic and Health Surveys division at ICF International through the completion of an online request form that is available on the DHS website (http://www.dhsprogram.com) from where the data was downloaded. As such, no ethical reviews and approvals were required before or during the preparation of the present manuscript. Further, informed consent was not required during the preparation of this paper

because there was no interaction with human subjects during the preparation of this manu-script. The 2016 UDHS was implemented by the Uganda Bureau of Statistics [16].

# Results

## Distribution of participant characteristics

Table 1 shows the distribution of covariates across the exposure of interest namely, a husband is justified in beating his wife for specific reasons. Of the 10,394 respondents, 46.5% (4, 821) reported that a husband is justified in beating his wife for specific reasons. The majority of those who justified wife-beating were females (78.3%), those aged 20–24 years (20.9%), those with a primary level of education (65.8%), and those who were currently in a union or living with a partner (86.4%). Further, there were statistically significant differences in sex, age, level of education, wealth index, religion, and controlling behaviour.

**Covariate balance before and after PSM.** Overall, we matched 8,284 respondents in a ratio of 1:1, and Table 2 shows the distribution in covariates after PSM. Before matching, sys-tematic differences in the justification of wife-beating were observed in all the covariates (stan-dardized percentage bias before matching >5%), except the number of living children, wealth index, and religion. After PSM, all the covariates demonstrated no systematic difference across the justification of wife-beating. Since all the standardized percentage biases for the covariates were <5%, this signifies a good covariate balance. We also observed at least a 75% drop in pseudo-R-square value (0.057 to 0.014), which is a large decline suggesting a good covariate balance was achieved.

## Distribution of outcomes before and after matching PSM

In Table 3, we summarized the distribution of respondents' experiences of different forms of IPV (outcomes) by the justification of wife-beating (exposure). Overall, there was a statistically significant difference in the study outcomes, namely lifetime experiences of sexual, physical, and emotional IPV by the justification of wife-beating. Before PSM, 1,156 (24.0%) respondents who justified wife-beating had experienced sexual IPV; 2,126 (44.1%) had experienced physical IPV, and 2,262 (46.9%) had experienced emotional IPV. After PSM, the data show that 978 (23.6) respondents experienced sexual IPV, 1,767 (42.7) had experienced physical IPV and 1,934 (46.7) had experienced emotional IPV. This indicates that there are no major differences in the overall proportions of respondents' experiences of different forms of IPV before and after PSM.

## Effect of justification of wife-beating on experiences of IPV

Logistic regression estimates after PSM in Table 4 show that justification of wife-beating is associated with higher chances of experiencing different forms of IPV both for men and women. Specifically, both men and women respondents combined, are about 1.6 times more likely to experience sexual IPV (OR, 1.67; 95% CI, 1.49–1.87), emotional IPV (OR, 1.63; 95% CI, 1.49–1.78), and physical IPV (OR, 1.69; 95% CI, 1.54–1.86) for those who justify wife-beat-ing compared to those who do not.

The findings of the sub-group analysis are also presented in Table 4. Men who justified wife-beating were about twice as more likely to experience emotional (OR, 2.27; 95% CI, 1.69–3.04), and physical (OR, 1.92; 95% CI, 1.35–2.72) IPV when compared to those who did not justify wife-beating. Further, men who justify wife-beating are about 1.5 times more likely to experience sexual IPV (OR, 1.45; 95% CI, 0.92–2.29) compared to those who do not justify wife-beating. Similarly, women who justify wife-beating were about 1.5 times more likely to

**Table 1. Distribution of participant characteristics.**

| Variable | | Overall (n = 10,363) | Wife-beating is justified | | p-value |
|---|---|---|---|---|---|
| | | | No (n = 5,542) | Yes (n = 4,821) | |
| Sex | | | | | |
| | Male | 2,846 (27.5) | 1,799 (32.5) | 1,047 (21.7) | <0.001 |
| | Female | 7,517 (72.5) | 3,743 (67.5) | 3,774 (78.3) | |
| Age group | | | | | |
| | 15–19 | 557 (5.4) | 228 (4.1) | 329 (6.8) | <0.001 |
| | 20–24 | 1,945 (18.8) | 939 (16.9) | 1,006 (20.9) | |
| | 25–29 | 2,077 (20.0) | 1,126 (20.3) | 951 (19.7) | |
| | 30–34 | 2,068 (20.0) | 1,155 (20.8) | 913 (18.9) | |
| | 35–39 | 1,475 (14.2) | 800 (14.4) | 675 (14.0) | |
| | 40–44 | 1,161 (11.2) | 679 (12.3) | 482 (10.0) | |
| | 45–49 | 855 (8.3) | 469 (8.5) | 386 (8.0) | |
| | 50–54 | 225 (2.2) | 146 (2.6) | 79 (1.6) | |
| | Mean (SD) | 31.8 (8.7) | 32.4 (8.7) | 31.1 (8.8) | <0.001 |
| Level of education | | | | | |
| | No education | 1,242 (12.) | 547 (9.9) | 695 (14.4) | <0.001 |
| | Primary | 6,260 (60.4) | 3,086 (55.7) | 3,174 (65.8) | |
| | Secondary | 2,049 (19.8) | 1,270 (22.9) | 779 (16.2) | |
| | Higher | 812 (7.8) | 639 (11.5) | 173 (3.6) | |
| Marital status | | | | | |
| | Currently in union | 8,999 (86.8) | 4,833 (87.2) | 4,166 (86.4) | 0.230 |
| | Formerly in union | 1,364 (13.2) | 709 (12.8) | 655 (13.6) | |
| Number of living children | | | | | |
| | ≤2 | 3,800 (36.7) | 2,088 (37.7) | 1,712 (35.5) | 0.074 |
| | 3–5 | 4,130 (39.9) | 2,176 (39.3) | 1,954 (40.5) | |
| | ≥6 | 2,433 (23.5) | 1,278 (23.1) | 1,155 (24.0) | |
| | Mean (SD) | 3.8 (2.7) | 3.8 (2.7) | 3.8 (2.6) | 0.84 |
| Wealth index | | | | | |
| | Poorest | 2,469 (23.8) | 1,075 (19.4) | 1,394 (28.9) | <0.001 |
| | Poorer | 2,214 (21.4) | 1,077 (19.4) | 1,137 (23.6) | |
| | Middle | 2,002 (19.3) | 1,054 (19.0) | 948 (19.7) | |
| | Richer | 1,866 (18.0) | 1,108 (20.0) | 758 (15.7) | |
| | Richest | 1,812 (17.5) | 1,228 (22.2) | 584 (12.1) | |
| Religion | | | | | |
| | No religion | 24 (0.2) | 14 (0.3) | 10 (0.2) | <0.001 |
| | Anglican | 3,316 (32.0) | 1,854 (33.5) | 1,462 (30.3) | |
| | Catholic | 4,324 (41.7) | 2,139 (38.6) | 2,185 (45.3) | |
| | Muslim | 1,222 (11.8) | 668 (12.1) | 554 (11.5) | |
| | Seventh Day Adventist | 145 (1.4) | 104 (1.9) | 41 (0.9) | |
| | Pentecostal | 1,200 (11.6) | 673 (12.1) | 527 (10.9) | |
| | Others | 132 (1.3) | 90 (1.6) | 42 (0.9) | |
| Place of residence | | | | | |
| | Urban | 2,106 (20.3) | 1,326 (23.9) | 780 (16.2) | <0.001 |
| | Rural | 8,257 (79.7) | 4,216 (76.1) | 4,041 (83.8) | |
| Partner has a controlling behavior | | | | | |
| | No | 2,822 (27.2) | 1,782 (32.2) | 1,040 (21.6) | <0.001 |
| | Yes | 7,541 (72.8) | 3,760 (67.8) | 3,781 (78.4) | |

*(Continued)*

**Table 1.** (Continued)

| Variable | | Overall (n = 10,363) | Wife-beating is justified | | p-value |
|---|---|---|---|---|---|
| | | | No (n = 5,542) | Yes (n = 4,821) | |
| Region | | | | | |
| | Kampala | 527 (5.1) | 373 (6.7) | 154 (3.2) | <0.001 |
| | Central1 | 863 (8.3) | 500 (9.0) | 363 (7.5) | |
| | Central2 | 850 (8.2) | 449 (8.1) | 401 (8.3) | |
| | Busoga | 889 (8.6) | 544 (9.8) | 345 (7.2) | |
| | Bukedi | 662 (6.4) | 280 (5.1) | 382 (7.9) | |
| | Bugishu | 596 (5.8) | 300 5.4) | 296 (6.1) | |
| | Teso | 668 (6.4) | 266 (4.8) | 402 (8.3) | |
| | Karamoja | 427 (4.1) | 134 (2.4) | 293 (6.1) | |
| | Lango | 734 (7.1) | 333 (6.0) | 401 (8.3) | |
| | Acholi | 654 (6.3) | 300 (5.4) | 354 (7.3) | |
| | West Nile | 727 (7.0) | 313 (5.6) | 414 (8.6) | |
| | Bunyoro | 675 (6.5) | 472 (8.5) | 203 (4.2) | |
| | Tooro | 722 (7.0) | 424 (7.7) | 298 (6.2) | |
| | Ankole | 802 (7.7) | 456 (8.2) | 346 (7.2) | |
| | Kigezi | 567 (5.5) | 398 (7.2) | 169 (3.5) | |

experience all three forms of IPV: sexual (OR, 1.67; 95% CI, 1.46–1.91), emotional (OR, 1.42; 95% CI, 1.27–1.60), and physical (OR, 1.65; 95% CI, 1.46–1.85), when compared to those who did not justify wife-beating.

**Sensitivity analysis.** The Mantel-Haenzel bounds analysis showed that a Gamma value of 1.55 was required for a shift from a statistically significant value to a statistically non-significant value. Since a large Gamma value was required to attain statistical non-significance in the Mantel-Haenzel bounds, the implication is that the findings are robust to unmeasured confounders and analytic approaches.

## Discussions

This study aimed to investigate the causal effects of the acceptability of gender norms justifying wife-beating on the life-time experiences of sexual, emotional, or physical IPV among Ugandan men and women using a quasi-experimental design with propensity-score matched (PSM) analysis. We analyzed data from the 2016 Uganda Demographic and Health Survey (UDHS) data. Intimate partner violence (IPV), which is violence that occurs between people in sexual or romantic relationships, harms the victim's physical, sexual or psychological health [4–6, 30].

Results show that the acceptance of gender norms justifying wife-beating increases the likelihood of experiencing sexual, emotional, and physical IPV in the general population–both men and women. This finding is not very different from that obtained from other similar studies. In one study, an analysis of IPV data from 30 African countries, it was reported that the incidence of IPV and the acceptance of wife-beating vary synchronously across both time and space, that is, communities and periods with lower levels of acceptance of wife-beating had lower levels of IPV incidence and vice versa [31]. In another study, results of a simulation experiment showed that communities that disapprove of wife-beating as a social norm had statistically significant reductions in the incidence of IPV at the population level, much as the effect was rather small with the incidence of any form of IPV falling by about 10 percentage points [3].

**Table 2. Covariate balance before and after PSM.**

| Variable | | Mean | | SPB* |
|---|---|---|---|---|
| | | **Treated** | **Control** | |
| Sex | | | | |
| | Unmatched | 0.78 | 0.68 | 24.4 |
| | Matched | 0.75 | 0.75 | -0.1 |
| Age in 5-year groups | | | | |
| | Unmatched | 3.83 | 4.08 | -14.6 |
| | Matched | 3.94 | 3.91 | 2.1 |
| Level of education | | | | |
| | Unmatched | 31.13 | 32.40 | -14.6 |
| | Matched | 31.71 | 31.53 | 2.1 |
| Marital status | | | | |
| | Unmatched | 1.09 | 1.36 | -36.7 |
| | Matched | 1.18 | 1.16 | 3.5 |
| Number of living children | | | | |
| | Unmatched | 1.14 | 1.13 | 2.3 |
| | Matched | 1.14 | 1.13 | 0.9 |
| Wealth index | | | | |
| | Unmatched | 1.88 | 1.85 | 4 |
| | Matched | 1.88 | 1.87 | 1.7 |
| Religious affiliation | | | | |
| | Unmatched | 3.81 | 3.80 | 0.4 |
| | Matched | 3.84 | 3.79 | 1.9 |
| Place of residence | | | | |
| | Unmatched | 2.59 | 3.06 | -34 |
| | Matched | 2.79 | 2.74 | 3.8 |
| Region | | | | |
| | Unmatched | 2.31 | 2.39 | -5 |
| | Matched | 2.37 | 2.35 | 1.1 |
| Partner has a controlling behavior | | | | |
| | Unmatched | 1.84 | 1.76 | 19.4 |
| | Matched | 1.82 | 1.83 | -1.9 |

**Note**: Standardized percentage bias (SPB) <5% signifies balance in covariate.

In the sub-group analysis, women who accepted gender norms justifying wife-beating experienced all three forms of IPV: sexual, emotional, and physical violence when compared to those women who did not justify wife-beating. For men, justification of wife-beating was associated with an increased likelihood of experiencing emotional and physical violence when compared to those men who did not justify wife-beating. However, whereas the odds of experiencing sexual IPV was higher among men justifying wife-beating compared to those who did not justify, the difference was not statistically significant. The finding of an increased odds of experiencing of sexual, emotional, and physical IPV among people who accept gender norms justifying wife-beating might be explained by two plausible reasons. First, gender norms are acquired through socialization during the transition from childhood to adulthood. Recent studies conducted in Uganda show that tendencies to have inequitable gender attitudes including justification of wife-beating can be traced in early adolescence [32] and tend to increase with age [33]. Studies have also report that people who experience violence in

**Table 3. Number of respondents (%) reporting experiences of IPV before and after matching PSM.**

| Experiences of IPV | | Before PSM | | | | After PSM | | | |
|---|---|---|---|---|---|---|---|---|---|
| | | Overall (n = 10,363) | Justify wife-beating | | | Overall (n = 8,284) | Justify wife-beating | | |
| | | | No (n = 5,542) | Yes (n = 4,821) | P-value | | No (n = 4,142) | Yes (n = 4,142) | P-value |
| Sexual IPV | | | | | | | | | |
| | No | 8,422 (81.3) | 4,757 (85.8) | 3,665 (76.0) | <0.001 | 6,667 (80.5) | 3,501 (84.5) | 3,164 (76.4) | <0.001 |
| | Yes | 1,941 (18.7) | 785 (14.2) | 1,156 (24.0) | | 1,617 (19.5) | 641 (15.5) | 978 (23.6) | |
| Physical IPV | | | | | | | | | |
| | No | 6,655 (64.2) | 3,960 (71.5) | 2,695 (55.9) | <0.001 | 5,222 (63.0) | 2,856 (69.0) | 2,375 (57.3) | <0.001 |
| | Yes | 3,708 (35.8) | 1,582 (28.5) | 2,126 (44.1) | | 3,062 (37.0) | 1,286 (31.0) | 1,767 (42.7) | |
| Emotional IPV | | | | | | | | | |
| | No | 6,182 (59.7) | 3,623 (65.4) | 2,559 (53.1) | <0.001 | 4,883 (58.9) | 2,687 (64.9) | 2,208 (53.3) | <0.001 |
| | Yes | 4,181 (40.3) | 1,919 (34.6) | 2,262 (46.9) | | 3,401 (41.1) | 1,455 (35.1) | 1,934 (46.7) | |

childhood find it normal to accept violence as a disciplinary measure and tend to become either victims or perpetrators of gender based violence, or both. The effects of childhood experiences of gender-based violence suffices in adulthood especially in marriage [34, 35]. Second, cultural beliefs or societal justification of gender-based violence even exacerbates the problem. For instance, certain cultures justify dating and marital rape while in other cultural settings, intimate love is intricately linked to legitimization of IPV [36, 37]. In addition, certain women feel loved if they are either abused or beaten by their spouses as a corrective measure for their faults. In addition, patriarchal norms that support male dominance and power control may further drive IPV [38].

Results further showed that women who justify wife-beating experience all three forms of violence compared to those who do not justify wife-beating. Conversely, men who justify wife-

**Table 4. Odds ratio estimates of the effect of justification of wife-beating on experiences of different forms of IPV.**

| Experiences of IPV | | Odds Ratio (95% CI) |
|---|---|---|
| **Both men and women combined** | | |
| Sexual violence (ref = No) | | 1.67 (1.49–1.87)** * |
| Emotional violence (ref = No) | | 1.63 (1.49–1.78)*** |
| Physical violence (ref = No) | | 1.69 (1.54–1.86)*** |
| **Subgroup analysis** | | |
| Sexual violence(ref = No) | | |
| | Males | 1.45 (0.92–2.29) |
| | Females | 1.67 (1.46–1.91)*** |
| Emotional violence (ref = No) | | |
| | Males | 2.27(1.69–3.04)*** |
| | Females | 1.42(1.27–1.60)*** |
| Physical violence(ref = No) | | |
| | Males | 1.92(1.35–2.72)*** |
| | Females | 1.65(1.46–1.85)*** |

**Note**: Statistical significance

* p<0.05

** p<0.01

*** p<0.001

beating experience merely physical and emotional violence but not sexual violence compared to those who do not justify wife-beating. This finding is in agreement with earlier studies in SSA that report IPV is more prevalent in women than men for various reasons, namely neglect of family responsibilities such as childcare, travelling away from home without informing a husband, or being disrespectful towards the spouse. Another likely explanation could be since sexual violence is largely perpetrated by men, in instances where it is perpetrated by women through for example denial of sex or forced sex, the men generally tend to not report for fear of shame and disempowerment or emasculation [39, 40].

Generally in most settings, being a man means being tough, brave, aggressive, and invulnerable, consequently, the need to appear invulnerable reduces men's willingness to seek help or treatment for physical or mental health problems, and in turn, this contributes to lower rates of safer sex and health-seeking behaviour [1]. Although evidence for the effectiveness of programs like SASA! to shift gendered norms, attitudes, and beliefs related to IPV is growing [11, 12], their potential to significantly impact the occurrence of IPV in a general population still requires rigorous evaluation for several reasons. Firstly, findings from these IPV prevention trials/programs primarily rely on self-reported changes in gendered social beliefs and norms among members of neighborhoods where the programs are being implemented. This measure can be strongly influenced by social desirability bias. Second, rates of justifying wife-beating have been used as indicators for the prevalence of IPV at the community level. However, changes in this measure at a community level do not inform the extent to which it translates to a reduction of IPV at the population level [3]. Understanding these facets is important to inform future interventions, national policy, and development partners about whether the investment is worthwhile or not, and whether such changes can translate to real reductions in IPV.

Some limitations in this study include reliance of the analytic technique on observed covariates but not unobserved (unmeasured) covariates and the lack of qualitative data to contextualize the findings. Further, the exposure and outcomes were assessed by self-reports implying that the possibility of social desirability bias cannot be excluded. However, a key strength of our study is that we analyzed a large and nationally representative dataset, implying that the findings are generalizable to the entire country and other similar settings. Secondly, we applied a robust study design and analytic approach to measure unbiased cause-effect relationships using observational data, implying that results are robust to unmeasured confounders and the analytic technique. Furthermore, the question items in 2016 UDHS tool, justification of wife-beating is premised on gender-norms that men should practice punishment acts on their wives as a form of corrective action when wives do not meet some expectations. As such, there remains an unanswered question on whether justification of wife-beating increases the risk for IPV in men or whether justification of wife beating is related to previous experience of IPV from the wife. We do have data to support this argument at the moment this is a knowledge gap that could be explored for other studies.

## Conclusions and recommendations

In conclusion, findings from this study show that the likelihood of experiencing sexual, emotional, and physical IPV is higher among both women and men who accept gender norms justifying wife-beating compared to those who do not. There is, therefore, a need for more research to study drivers for acceptance of gender norms justifying wife-beating. In this way, appropriate government agencies can be able put in place mechanisms to address the acceptability of gender norms justifying wife-beating at the societal level. For instance, it has been recommended elsewhere that strategies for reducing gendered social norms related IPV could

incorporate a wider range of individual, household, community, institutional, and societal factors that view violence in the family as unacceptable and that can be acted on in order to contribute to generational efforts to eliminate IPV [3]. One strategy could be for policy makers to repackage scientifically proven interventions such as the SASA! Program [11, 12] with a combination structured legal platforms for perpetrators, as well as intensive programs to educate men, women, boys and girls to adopt more prohibitive social norms against negative attitudes such as justification of wife-beating for any reason.

## Acknowledgments

We wish to acknowledge the Ministry of Health of Uganda and the Demographic Health Survey program, which granted us access to use the DHS data. This manuscript preparation and statistical data analysis were carried out, during an Impact Evaluation Training Workshop as part of the "Contextualizing IE Pedagogy in Africa (CIPA) project" a joint initiative between the Centre for Global Challenges based in Utrecht University (UU) and the Network of Impact Evaluation Researchers in Africa (NIERA) based at the United States International University (USIU-Africa).

## Author Contributions

**Conceptualization:** Damazo T. Kadengye, Jonathan Izudi, Elizabeth Kemigisha, Sylvia Kiwuwa-Muyingo.

**Data curation:** Damazo T. Kadengye, Jonathan Izudi.

**Formal analysis:** Damazo T. Kadengye, Jonathan Izudi, Sylvia Kiwuwa-Muyingo.

**Investigation:** Damazo T. Kadengye.

**Methodology:** Damazo T. Kadengye, Jonathan Izudi, Elizabeth Kemigisha.

**Validation:** Damazo T. Kadengye, Jonathan Izudi, Sylvia Kiwuwa-Muyingo.

**Visualization:** Damazo T. Kadengye, Jonathan Izudi.

**Writing – original draft:** Damazo T. Kadengye, Jonathan Izudi, Elizabeth Kemigisha, Sylvia Kiwuwa-Muyingo.

**Writing – review & editing:** Damazo T. Kadengye, Jonathan Izudi, Elizabeth Kemigisha, Sylvia Kiwuwa-Muyingo.

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
