## [Decision Letter · Decision Letter 0]

12 Dec 2022

PONE-D-22-26659Effect of justification of wife-beating on experiences of intimate-partner violence among men and women in Uganda: A propensity matched scores approachPLOS ONE

Dear Dr. Kadengye,

Thank you for submitting your manuscript to PLOS ONE. After careful consideration, we feel that it has merit but does not fully meet PLOS ONE’s publication criteria as it currently stands. Therefore, we invite you to submit a revised version of the manuscript that addresses the points raised during the review process.

We look forward to receiving your revised manuscript.

Kind regards,

Joseph KB Matovu, Ph.D.

Academic Editor

PLOS ONE

Journal Requirements:

Reviewers' comments:

Reviewer's Responses to Questions

**Comments to the Author**

1. Is the manuscript technically sound, and do the data support the conclusions?

Reviewer #1: Partly

Reviewer #2: Yes

2. Has the statistical analysis been performed appropriately and rigorously? 

Reviewer #1: Yes

Reviewer #2: Yes

3. Have the authors made all data underlying the findings in their manuscript fully available?

Reviewer #1: Yes

Reviewer #2: Yes

4. Is the manuscript presented in an intelligible fashion and written in standard English?

Reviewer #1: No

Reviewer #2: Yes

5. Review Comments to the Author

Reviewer #1: Comments

First I would like to thank the editors for inviting me this interesting paper and the authors for coming up with this novel work.

Abstract

1. Why not you used structured abstract form (kindly write the parts of an abstract in a separate line).

2. “…the majority (3,774; 78.3%) were women compared to men (1,047; 21.7%)” with regard to what, I think something is missed in this sentence.

3. “We found that at population level…” how do you address multilevel factors by using an ordinary logistic regression?

4. “Government should tackle the drivers of acceptability of gender norms justifying wife beating at the societal level.” What are the drivers of acceptability of gender norms justifying wife beating? Do you obtain those drivers In your study? Be specific with your recommendations?

Introduction

5. Para 1, line 2-5 is too long. Please paraphrase it.

6. Please try to show the problem, gaps and tried solutions to reduce IPV globally, in SSA,,,, Uganda.

7. Line 13-16 needs a reference.

8. Line 15-6, “several programmes aimed at shifting norms and behavior around IPV have been implemented across Africa” what types of programmes were implemented? What was their outcome? What type of norms and behaviors they intended to change? Please explain with examples.

9. Please write the long version of “SASA! “?

10. Line 20, needs a reference.

11. Line 24 needs a reference.

12. Line 26 needs a reference.

Methods and Materials

13. How do use quasi experimental design for secondary data? What does experimental study design mean? What is the difference between experimental and retrospective study designs? In quasi experimental design you are going to assign the subjects into control and study/intervention group then you will measure your outcome variable both at the base line, and end of your intervention, please make it clear in the methods part specifically in exposure part.

14. What does mean gender norm? What type of norms (positive, negative) are you going to assess as a means of justification?

15. Put references to your outcome variables (line 104-112)

16. Line 114-124, where and how do you get the independent variables?

17. Why you measure life time prevalence rather 12months prevalence of IPV? Which one do you think more important for public health intervention?

18. Why do not you compute the composite variable of IPV?

19. How do you pair each pre-test score to the corresponding post-test score? How do you obtain your post-intervention/test score? How do you prefer retrospective cohort over quasi experimental design for your study? You need to clarify it

20. How do you control the effect of other covariates on IPV? How do you know the real effect of your independent variable to the outcome variable?

21. why you are interested to study the effect of acceptability of gender norms justifying wife-beating on the life-time experiences of sexual, emotional, or physical intimate partner violence (IPV)? Why not other variables? Why you are interested to study among the two groups (men and women)?

Conclusions and Discussions

22. Please mention them separately. Your conclusion should follow the discussion.

23. Your discussion seems result. it needs further clarification and comparison with other studies. Please rewrite and try to widen your discussion according to your findings.

24. Line 230 “Gender-based violence (GBV) of any form perpetrated against a woman or girl, man or 231 boy harms their physical, sexual or psychological health”. Why you mentioned GBV in this section? Do you think IPV and GBV are similar?

25. “In this paper, we investigated the causal effects of acceptability of gender norms justifying wife-beating on the life-time experiences of sexual, emotional, or physical intimate partner violence (IPV) among Ugandan men and women using a quasi-experimental design with propensity-score matched (PSM) analysis. We analyzed 235 data from the 2016 Uganda Demographic and Health Survey (UDHS) data” this sentence should come at the beginning of your discussion, so avid the sentence about GBV preceding it. Then rewrite it as “the aim of this study was to investigate the causal effects of acceptability of gender norms justifying wife-beating on the life-time experiences of sexual, emotional, or physical intimate partner violence (IPV) among Ugandan men and women using 2016 Uganda Demographic and Health Survey (UDHS) data.”

26. “, the exposure and outcomes were assessed by self-reports implying that the possibility of social desirability bias cannot be excluded” what was your effort to reduce this bias?

27. “In conclusion, findings from this study show that the risk of experiencing IPV is high…..” how is your finding high? Compared to what?

28. “…..There is, therefore, a need to tackle the drivers of gender norms justifying wife-beating at the societal level.” What are the driving factors? Your recommendation seems shallow please rewrite it and be specific with your finding? (what to do, how to do, who should do it)?

29. Do you think you answered your research question? What was your hypothesis?

30. You have to acknowledge your study participants.

General comments

31. How long do you follow your study subjects to measure the post-test score? What is your level of blinding?

32. Please put a detailed description about your intervention?

33. Intervention and control group?

34. Outcome and exposure ascertainment issue is questionable?

35. Baseline data, and intervention data is needed.

36. The average or percentage of difference across groups in terms of outcome and exposure?

Good Luck!!!

Reviewer #2: Title

This paper makes important contribution to the literature on intimate partner violence in Uganda. Just a minor suggestion is that it should reflect the timeline of study. For example, A propensity matched scores approach on 2016 UDHS or something like that.

Another thing is that it is vague whether the study was carried out among both married and never-married people. If not among never-married people, the title should be specified accordingly. For example, married men and women or couples.

Introduction

This session highlighted the linkage between justification of gender norms and the risk of IPV. One unclear thing is that the concept between justification of wife beating and the risk of IPV in men. Does it mean that a man who justified wife-beating has more experienced with IPV by his wife? Generally, justification of wife-beating is based on gender-norms that men should practice punishment acts on their wives. So, if a man justifies wife-beating, he will probably more perform IPV on his wife. Or does this area remain as a knowledge gap?

Research aim

The research aim well reflects the purpose of this study.

Methodology

The use of propensity-score matched (PSM) analysis on a cross-sectional data is appropriate for this study that investigates causal effects. The sampling method is clear. However, is there any inclusion criteria or exclusion criteria for asking the question – justification of wife-beating? Was this question asked to every participant even if the participant was never married one? Then the responses were included in the analysis. If so, how it linked with IPV? According to Line 102-103, “[IPV] was measured by asking all ever-married women and men if their spouse/partner ever did the following to them”.

Regarding the covariates, does the partner’s alcohol consumption not have effect on experience of IPV?

Statistical Method

The use of STATA software is appropriate for this study.

Detail explanation helped the reader in understanding the statistical procedure.

Results

The finding represents the statistical analysis performed by the authors.

Discussion

This study well discussed about the findings by using comparative global studies and sound theoretical background.

Just a curiosity, why did the analysis not include the childhood experience of gender-based violence (e.g. Witnessing parental violence) as a covariate?

Conclusion

The authors’ recommendations are appropriate, but it would be more effective if the conclusion session is separately written from discussion session. If available, it should suggest some interventions that could reduce gender norms justifying wife beating at the societal level. Again, the concept for the increased risk of IPV in men who accepts wife-beating norms is still unclear. If this study does not explain, it should be suggested for future studies.

Grammatical errors

Please review the manuscript for grammatical and spelling errors.

Minor error – [Line 240] Please revise the phrase (double justify words): “who did not justify justifying wife-beating”

6. PLOS authors have the option to publish the peer review history of their article (what does this mean?). If published, this will include your full peer review and any attached files.

Reviewer #1: No

Reviewer #2: No

---

## [Author Response · Author response to Decision Letter 0]

26 Jan 2023

We appreciate the two reviewers who provided us with constructive comments that have helped us improve the manuscript.

We have provided our comments to each and every raised comments. These are included in the document "Response to Reviewers". All the comments were very helpful and we have reflected all the changes in the Manuscript with Track Changes.

We look forward to the publication of this important paper in PLOS ONE.

---

## [Decision Letter · Decision Letter 1]

28 Mar 2023

Effect of justification of wife-beating on experiences of intimate partner violence among men and women in Uganda: A propensity-score matched analysis of the 2016 Demographic Health Survey Data

PONE-D-22-26659R1

Dear Dr. Kadengye,

We’re pleased to inform you that your manuscript has been judged scientifically suitable for publication and will be formally accepted for publication once it meets all outstanding technical requirements.

Kind regards,

Joseph KB Matovu, Ph.D.

Academic Editor

PLOS ONE

Additional Editor Comments (optional):

Reviewers' comments:

Reviewer's Responses to Questions

**Comments to the Author**

1. If the authors have adequately addressed your comments raised in a previous round of review and you feel that this manuscript is now acceptable for publication, you may indicate that here to bypass the “Comments to the Author” section, enter your conflict of interest statement in the “Confidential to Editor” section, and submit your "Accept" recommendation.

Reviewer #1: All comments have been addressed

Reviewer #2: All comments have been addressed

2. Is the manuscript technically sound, and do the data support the conclusions?

Reviewer #1: Yes

Reviewer #2: Yes

3. Has the statistical analysis been performed appropriately and rigorously? 

Reviewer #1: Yes

Reviewer #2: Yes

4. Have the authors made all data underlying the findings in their manuscript fully available?

Reviewer #1: Yes

Reviewer #2: Yes

5. Is the manuscript presented in an intelligible fashion and written in standard English?

Reviewer #1: Yes

Reviewer #2: Yes

6. Review Comments to the Author

Reviewer #1: thank you very much the authors for addressing all my concerns and wish you all the best in publishing your paper.

Reviewer #2: Minor suggestion:

Line 314 - 316 "We do have data to support this argument at the moment this is a knowledge gap that could be explored for other studies".

This sentence seems contradictory between "available data" and "knowledge gap".

Please clarify whether "we do not have" or "we do have".

7. PLOS authors have the option to publish the peer review history of their article (what does this mean?). If published, this will include your full peer review and any attached files.

Reviewer #1: No

Reviewer #2: No

---

## [Editor Report · Acceptance letter]

3 Apr 2023

PONE-D-22-26659R1 

Effect of justification of wife-beating on experiences of intimate partner violence among men and women in Uganda: A propensity-score matched analysis of the 2016 Demographic Health Survey Data 

Dear Dr. Kadengye:

I'm pleased to inform you that your manuscript has been deemed suitable for publication in PLOS ONE. Congratulations! Your manuscript is now with our production department. 

Kind regards, 

on behalf of

Dr. Joseph KB Matovu 

Academic Editor

PLOS ONE